# Machine Learning for Detecting Total Knee Arthroplasty Implant Loosening on Plain Radiographs

**DOI:** 10.3390/bioengineering10060632

**Published:** 2023-05-23

**Authors:** Man-Soo Kim, Ryu-Kyoung Cho, Sung-Cheol Yang, Jae-Hyeong Hur, Yong In

**Affiliations:** Department of Orthopaedic Surgery, Seoul St. Mary’s Hospital, College of Medicine, The Catholic University of Korea, 222, Banpo-daero, Seocho-gu, Seoul 06591, Republic of Korea; kms3779@naver.com (M.-S.K.); dreamer1222@naver.com (R.-K.C.); scillay@gmail.com (S.-C.Y.); rosebulls96@gmail.com (J.-H.H.)

**Keywords:** loosening, total knee arthroplasty, radiograph, machine learning, convolutional neural network, transfer learning

## Abstract

(1) Background: The purpose of this study was to investigate whether the loosening of total knee arthroplasty (TKA) implants could be detected accurately on plain radiographs using a deep convolution neural network (CNN). (2) Methods: We analyzed data for 100 patients who underwent revision TKA due to prosthetic loosening at a single institution from 2012 to 2020. We extracted 100 patients who underwent primary TKA without loosening through a propensity score, matching for age, gender, body mass index, operation side, and American Society of Anesthesiologists class. Transfer learning was used to prepare a detection model using a pre-trained Visual Geometry Group (VGG) 19. For transfer learning, two methods were used. First, the fully connected layer was removed, and a new fully connected layer was added to construct a new model. The convolutional layer was frozen without training, and only the fully connected layer was trained (transfer learning model 1). Second, a new model was constructed by adding a fully connected layer and varying the range of freezing for the convolutional layer (transfer learning model 2). (3) Results: The transfer learning model 1 gradually increased in accuracy and ultimately reached 87.5%. After processing through the confusion matrix, the sensitivity was 90% and the specificity was 100%. Transfer learning model 2, which was trained on the convolutional layer, gradually increased in accuracy and ultimately reached 97.5%, which represented a better improvement than for model 1. Processing through the confusion matrix affirmed that the sensitivity was 100% and the specificity was 97.5%. (4) Conclusions: The CNN algorithm, through transfer learning, shows high accuracy for detecting the loosening of TKA implants on plain radiographs.

## 1. Introduction

Total knee arthroplasty (TKA) is a validated and effective surgical treatment for the functional recovery of and significant improvements in the quality of life of patients with end-stage knee osteoarthritis (OA) [1,2,3,4,5,6,7,8]. TKA is performed in over 1 million cases in the United States annually and is expected to continue to increase in frequency [9]. TKA has been shown to have a 15-year survival rate of 95% or greater due to the continuous development of technology [10]. However, due to increases in the frequency of arthroplasty and in life expectancy, the revision burden is also expected to continue to increase [9,11,12].

TKA failure may occur due to various causes, the most common being periprosthetic loosening [13,14]. Precise detection of prosthetic loosening of TKA implants is challenging even for experienced clinicians and may not always be feasible, potentially delaying diagnosis. Multiple imaging tools are used for diagnosis, including plain radiographs, scintigraphy, arthrograms, fluorodeoxyglucose-positron emission tomography (FDG-PET), and magnetic resonance imaging (MRI) [15,16]. However, except for plain radiographs, the available imaging tools are invasive and expensive and do not actually confer significant benefits in terms of cost effectiveness compared to plain radiographs [17]. The concordance rate also varies as readings are often performed by clinicians who are not experts [18,19].

Due to recent developments, machine learning has been widely used in orthopedic surgery. Especially in the field of artificial joints, machine learning is being studied a lot. For patients, it is used as a decision aid for diagnosis, the severity and complexity of OA, and the appropriate indication of artificial joint surgery, and machine learning is also used for implant size, positioning or alignment, and ligament balancing. In particular, machine learning is showing excellent performance in diagnosis through images [20,21]. Machine learning techniques are used to diagnose lung disease and breast cancer on radiographs and to support diagnoses in hospitals [22,23]. Machine learning can be used to detect prosthetic loosening after arthroplasty in orthopedic surgery, but the studies are still lacking [24,25,26,27,28]. Therefore, the purpose of this study was to investigate whether the prosthetic loosening of TKA implants could be detected precisely on plain radiographs using a deep convolution neural network (CNN).

## 2. Materials and Methods

We included patients who underwent revision TKA with prosthetic loosening at a single institution from 2012 to 2020. After obtaining Institutional Review Board (IRB) approval, we collected image data for patients diagnosed with prosthetic loosening after revision TKA during the study period. Preoperative radiographs of arthroplasty surgery for each patient were also collected. In revision TKA, anteroposterior (AP) radiographs were collected at the original resolution.

Images of a total of 100 revision TKA patients were collected and categorized as “loosened”. Cases classified as loosened were defined only when revision TKA was performed because TKA implant loosening was demonstrated in the intraoperative field as well as preoperatively. In order to collect information on well-fixed TKA for comparison, we conducted a survey of primary TKA patients during this period and performed propensity score matching (one-to-one) to minimize selection bias. One hundred patients who underwent primary TKA without loosening were extracted through propensity score matching for age, gender, body mass index, operation side, and American Society of Anesthesiologists (ASA) class. These patients were labeled as “fixed”.

In order to better visualize loosening around the knee implant, we obtained the original X-ray image of each TKA including only the knee implant periphery. The image was separated using the cropping technique through numpy in Python by focusing the image around the implant–bone interface, which was recognized as the most important position, to better confirm loosening around the implant. We compared AP knee implant X-ray images of loosened and fixed groups after resizing the images to 224 × 224 pixels. Since the size of the original image was different for each image, it was modified and there was no significant change from the original image. The image data set was too small to train the deep learning model, so we used an augmentation technique to balance the training image classes and increase the size of the training set to avoid overfitting. Rotation, width shift, height shift, zoom, and flip techniques were used. Rotation was allowed up to 360 degrees, and the width, height shift, and zoom range were set to 0.2 [29,30]. Transfer learning was used to create a model that detects TKA loosening. Transfer learning is a strategy for transferring the knowledge extracted by a neural network from specific data to solve a problem and applying it to a new task, including new and usually insufficient data to train neural networks from the outset [31,32].

In this study, we used the CNN model VGG 19 net, which is widely used in image classification algorithms because it has an easy-to-use structure and good performance [32]. VGG19, proposed by Simonyan and Zisserman [32], consists of 16 convolutional layers and 3 fully connected 19 layers to classify images into 1000 object categories. The model was trained using approximately 1.2 million images, including 100,000 images for testing and 50,000 images for validation. This is a very popular method for image classification because it uses multiple 3 × 3 filters for each convolutional layer. The architecture of VGG19 includes 16 convolutional layers for feature extraction and 3 layers for classification. The layers used for feature extraction are divided into 5 groups, with max pooling layers following each set. A 224 × 224 pixel image is input to this model, and the model outputs the labels of the objects in the image [32] (Figure 1).

We performed transfer learning with fine-tuning to reuse the model and improve accuracy. We implemented the VGG 19 CNN algorithm by replacing the fully connected layer (classifier) with 7 layers as follows: global average pooling, batch normalization, dropout (0.5), fully connected neural network (dense layer) with 256, batch normalization, dropout (0.5) layer to further reduce overfitting, and 1 output neuron for binary classification. For transfer learning models, cross-entropy was used as the loss function, and rectified linear unit (ReLU) was used as the activation function. The final output node used the Softmax activation function to classify the highest output for each class as the correct answer class [33,34,35,36].

The performance of the algorithm was analyzed while gradually changing the freezing part of the end layer. We used two methods for transfer learning. First, the fully connected layer was removed, and a new fully connected layer was added to construct a new model. The convolutional layer was frozen without training, and only the fully connected layer was trained (transfer learning model 1). Second, a new model was constructed by adding a fully connected layer and varying the range of freezing for the convolutional layer (transfer learning model 1). In this study, a fine-tuned model was designed by freezing up to convolutional block 4 and re-training from convolutional block 5 to the new fully connected layer [33,34,35,36] (Figure 2).

Patients in our study were divided into training (80%) and test (20%) groups. The model was trained on training patients and tested on test patients. The best models were identified by patient-rated performance. We then evaluated test patients using this best model to determine performance with an independent data set.

## 3. Statistical Analysis

Data were analyzed for both groups and are presented as mean and standard deviation. Chi-square tests were applied to identify significant differences between categorical variables, and we used the Wilcoxon signed-rank test to assess the significance of differences between continuous variables. In the case of propensity score matching, fixed cases were four times more than loose cases, so fixed cases could be adjusted 1:1 according to loosened cases. For each subsequent change to the model, starting with the first model tuned by transfer learning with fine-tuning of the CNN architecture, the model was tested on a test patient. Accuracy in this context is defined as the agreement of the model predictions to known values defined as intraoperative findings of “fixed” or “loosened” implants. Statistical analysis was performed using SPSS^®^ for Windows v21.0, with *p* < 0.05 indicating statistical significance.

## 4. Results

A total of 126 patients underwent revision TKA from 2010 to 2020. Twenty-six patients were excluded, including seven due to polyethylene wear, three due to fracture, and sixteen due to infection, with no loosening confirmed. In 100 patients, implant loosening of tibia or femur was confirmed on imaging and surgical records. The average age of the 100 patients was 70.4 years, and 80.0% were female. The average BMI was 26.3, and 37.0% underwent revision TKA on the left side. A control pool was recruited, including 399 patients who underwent primary TKA in 2020, and a control group of 100 patients was extracted through propensity score matching. In the control group, the average age was 70.9 years, 80.0% were female, the average BMI was 26.5, and 37.0% underwent revision TKA on the left side. There were no differences in demographic data between the two groups (all *p* > 0.05) (Table 1). Before propensity score matching, there was a significant difference in gender and operation side between the two groups, but after propensity score matching, both gender and operation side showed the same results between both groups.

Using 500 epochs, the transfer learning model 1 gradually increased in accuracy and ultimately reached 87.5%. The loss rate gradually decreased to 0.2527 (Figure 3). After assessing the results in the validation set through the confidence matrix, the accuracy was 87.5%, the sensitivity was 90%, the specificity was 100%, the positive predictive value was 100%, and the negative predictive value was 80% (Table 2).

Transfer learning model 2, which was trained on the convolutional layer, gradually increased in accuracy and reached 97.5%, which was a larger improvement than with model 1. The loss rate was 0.0307, confirming improvement compared to model 1 (Figure 4). After checking the results in the validation set through the confidence matrix, the accuracy was 97.5%, the sensitivity was 100%, the specificity was 95%, the positive predictive value was 95.2%, and the negative predictive value was 100%. Transfer learning model 2 exhibited a better sensitivity to detect loosening cases compared to model 1 (Table 2, Figure 5).

Actual fixed and loosened cases were evaluated using transfer learning model 2, and we confirmed that they were accurately judged (Figure 6).

## 5. Discussion

Because the final diagnosis of prosthetic loosening of TKA implants remains challenging, particularly in the early stages [17], interest in the use of artificial intelligence-based algorithms as a diagnostic tool is increasing [24,25,26,27,28]. The accurate detection of prosthetic loosening of TKA is difficult even for experienced surgeons [18,19]. In this study, the accuracy of detection for prosthetic loosening of TKA was confirmed using a transfer learning model based on VGG 19, a pre-trained CNN model. Two transfer learning model techniques were used with different ranges of freezing, and 97.5% accuracy was achieved using only images with fine-tuning.

The main purpose of TKA is to relieve the patient’s pain and restore function for a long time [37]. It is well known as the most effective and satisfactory surgical treatment in the field of orthopedic surgery and has a very good survival rate [38,39,40,41,42]. The 15-year survival rate is 93.0%; the 20-year survival rate is 90.1%; and the 25-year survival rate is reported to be 82.3% [37]. The result of such excellent long-term survival rates was made possible by the development of artificial joint materials and continuous technology [39]. Unfortunately, artificial joints always have a risk of failure due to various reasons [14]. There are various causes of TKA failure, including loosening, infection, instability, and persistent pain. Among them, the most common cause of TKA failure is loosening caused by osteolysis [14]. Demand for detecting implant loosening has been continuously made [43]. However, it is still difficult to detect implant loosening even though various methods have been used. Classen et al. analyzed aseptic loosening of TKA by bone scintigraphy and reported a sensitivity of 76%, specificity of 83%, positive predictive value of 93%, and a negative predictive value of 56% [43]. Sterner et al. evaluated the loosening of TKA using positron emission tomography and showed a large difference with sensitivity 100% and specificity 56% [44]. Mayer Wagner et al. reported that the detection of TKA loosening using positron emission tomography was less accurate than the detection of THA loosening [45]. The detection of TKA loosening by positron emission tomography showed a sensitivity of 56% for aseptic loosening and 14% for septic loosening [45]. As such, it is difficult to detect TKA loosening, even with more advanced imaging techniques than plain radiographs, and it is common to show large differences in sensitivity, specificity, and accuracy [43,44,45]. In diagnosing implant loosening, many studies have been conducted on many expensive imaging techniques, including positron emission tomography or bone scintigraphy [43,44,45]. However, when compared with a simple radiograph, it did not show a significant improvement in its diagnostic accuracy [43,44,45]. As an alternative to these limitations, interest in diagnostic techniques that have constant and relatively high accuracy in evaluating TKA loosening has increased, and machine learning is receiving great attention as an alternative [24,25,26,27,28]. Machine learning is an innovative method that can improve the diagnosis accuracy without being expensive, exposed to radiation, or invasive [24,25,26,28].

Among machine learning, deep learning is a more advanced and complex form of machine learning that mimics the neural connections in the brain using artificial neural networks (ANNs) arranged and organized in a hierarchical structure [46]. CNNs are another type of deep learning used in computer vision tasks, including medical image analysis, because they show excellent ability for image discrimination and processing [46]. Deep learning analysis technology for evaluating medical images is being applied to radiographic, computed tomography, ultrasound, MR, and fluoroscopic imaging, and significant results have been reported for the diagnosis of diseases of the chest, heart, brain, and breast [47]. In orthopedic surgery, image analysis studies using CNN algorithm structures have been conducted [48], such as measuring bone age using radiographic images [49] and recognizing fractures [50]. To improve the prediction accuracy of models, it is necessary to precede the construction with learning of high-quality images correctly classified by experts and a large training image dataset [51].

However, there are practical difficulties in securing data from large imaging sets. In addition, large-capacity learning data require a considerable learning time, even if high-performance hardware is used [35]. The method proposed to solve this problem is a transfer learning model, which increases the accuracy of the model by adding training data suitable for the study to an existing neural network trained with large amounts of more general data [52]. CNN-based algorithms such as Alexnet [53], VGGNet [32], GoogLeNet [54], and Residual Neural Network (ResNet) [55], proposed through the ImageNet image classification challenge competition, are used for transfer learning. VGGNet is a convolutional neural network with 16 deeply stacked convolution layers and is widely used as a deep learning transfer model in image analysis fields because of its high performance [32]. In this study, the VGG-19 deep learning algorithm, derived from VGGNet, was applied through transfer learning, and an accuracy greater than 95% was achieved.

In general, transfer learning using fine-tuning is a multi-step process. First, the fully connected layer at the end of the network is removed. Second, the fully connected layer is replaced with freshly initialized layers. Third, the earlier convolutional layers are frozen earlier in the network to ensure that previous robust features learned by the CNN are not destroyed. Model training is initiated only for the fully connected layer. Additionally, some/all of the convolutional layers in the network are unfrozen, and a second pass of training can be used to improve functionality [35,36,56]. If the size of the data set is large, it is recommended to use a transfer learning method that trains the entire model. However, the data set is not large in most cases when medical images are used, so most transfer learning models in the medical field only train the fully connected layer [57]. Shah et al. [28] used a transfer learning model for the detection of implant loosening in total hip arthroplasty (THA) and TKA, changing only the last linear layer representing the output with training performed for the entire model. This transfer learning method yielded an accuracy of about 80%. Lau et al. [25] used Xception, a pre-trained model for detecting TKA implant loosening, and observed greater than 96% accuracy through a transfer learning model that trains the whole structure. However, when a pre-trained model is used for medical images with small datasets and low data similarity, it is preferable to only train a part of the model [35,36,56]. In our study, only the fully connected layer was changed to a new fully connected layer, and one model that only trained the fully connected layer and another model that partially trained the convolutional layer were used. In VGG architecture, fine-tuning only the top block results in the best performance compared to the transfer learning model, which relearns the entire model [57]. Such tuning was used with a transfer learning model for detecting prosthetic loosening of TKA in this study for the first time. Our method of training up to the last convolutional layer resulted in the best accuracy.

Our results, including the accuracy, sensitivity, and specificity of evaluating TKA implant loosening using only X-ray images, are not inferior to those of other studies. Previously, Lau et al. [25] analyzed TKA loosening using an image-based machine learning model on 440 knee radiograph images, including 206 with loosening and 234 without loosening. They used the Xception model without fine-tuning and conducted 5000 epochs, resulting in an accuracy of 96.3%, a sensitivity of 96.1%, a specificity of 90.9%, a positive predictive value of 92.4%, a negative predictive value of 95.2%, and an AUC of 93.5%. The accuracy evaluated by the senior orthopedic specialist was 89.09–94.54%. Shah et al. [28] evaluated a loosening detection model using 217 fixed TKA and 137 loosened TKA X-rays. Among the CNN models, ResNet, AlexNet, Inception, and DenseNet were used. The image modification method was used for the transfer learning model, and changes in the layer were not performed. The resulting accuracy was 70.8% when using raw images and untrained models, 73.3% with segmentation and cropping tools, and greater than 80% when using pre-trained models using large datasets. When only the image was used, the accuracy was less than 90%; when clinical information was added, the accuracy was greater than 90%. Compared to THA, the accuracy of detecting TKA loosening was lower. In our study, the transfer learning model 2 showed 97.5% accuracy in the training set and greater than 95% accuracy, sensitivity, and specificity in the test set.

Our study has limitations. First, we only included patients who underwent revision TKA due to loosening. The model’s performance cannot be judged for those whose loosening has not yet been recognized because the model only judged the confirmed loosening of TKA implants on plain radiographs. Therefore, it is questionable whether our model would be effective for detecting loosening during the early stages. It is thought that it can be the basis for developing a model that can recognize loosening at an early stage. Second, preprocessing, which resizes and crops images to substitute for transfer learning models, can reduce image resolution, which can have a significant impact on the model’s performance. Third, since the patients were collected retrospectively, prospective X-ray follow-up of patients is necessary to further validate our results. Fourth, the pre-trained model used in this study was evaluated using only VGGNet [32]. There are several pre-trained models using Imagenets, and a more advanced prosthetic loosening detection model using transfer learning may be established through additional evaluation [28]. Fifth, image augmentation techniques or preprocessing are necessary to overcome reduced image resolution due to the size of images. This may affect performance, including the accuracy of the transfer learning model [29,30]. Sixth, as in other studies, additional patient demographic data and clinical information could further improve the accuracy of the prosthetic loosening model [25,28]. However, the purpose of this study was to detect loosening using only images. Seventh, most of the patients included in the study were women, as the majority of patients undergoing TKA in Asia are women [58,59,60,61]. Gender differences may affect the performance of the model, and further studies that include additional male patients are needed. Finally, the number of images used in this study was insufficient. In general, more than 1000 images are necessary to train a machine learning model [62]. To overcome this, we used image augmentation technology to achieve greater than 95% accuracy. However, it is necessary to include more image data in future studies.

## 6. Conclusions

The application of the CNN algorithm through transfer learning results in high accuracy for detecting prosthetic loosening of TKA implants by studying plain radiographs. This method may be utilized as an auxiliary tool for diagnosing prosthetic loosening of TKA implants, but it is judged that additional research is needed.

## Figures and Tables

**Figure 1 bioengineering-10-00632-f001:**
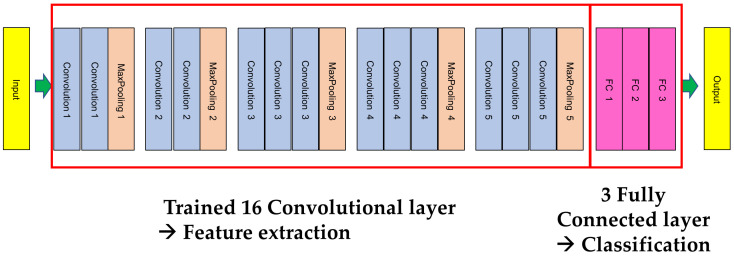
Visual Geometry Group (VGG) 19 architecture.

**Figure 2 bioengineering-10-00632-f002:**
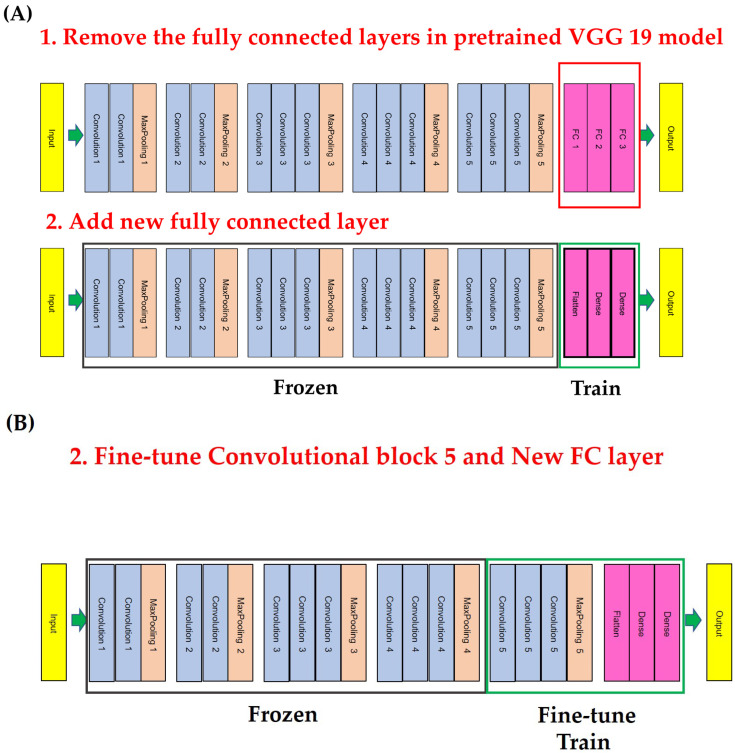
Different transfer learning techniques. (**A**) In the first technique, in which the source weights are fixed and the original fully connected layers would be replaced by new fully connected layers to suit the target dataset, the convolutional layer is frozen without training, and only the fully connected layer is trained. (**B**) In the second technique, a new model was constructed by adding the last fully connected layer and varying the range of freezing for the convolutional layer.

**Figure 3 bioengineering-10-00632-f003:**
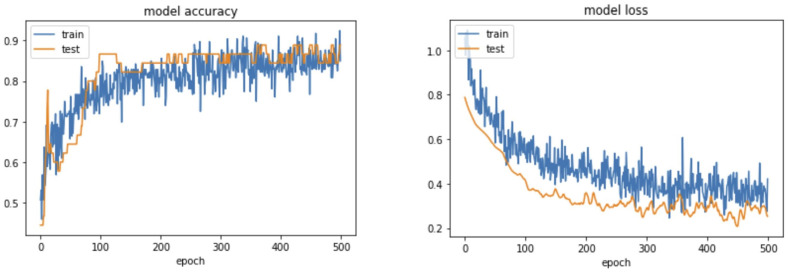
Accuracy and loss curves of transfer learning model 1.

**Figure 4 bioengineering-10-00632-f004:**
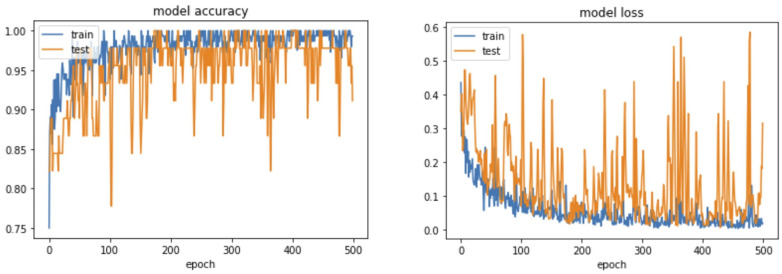
Accuracy and loss curves of transfer learning model 2.

**Figure 5 bioengineering-10-00632-f005:**
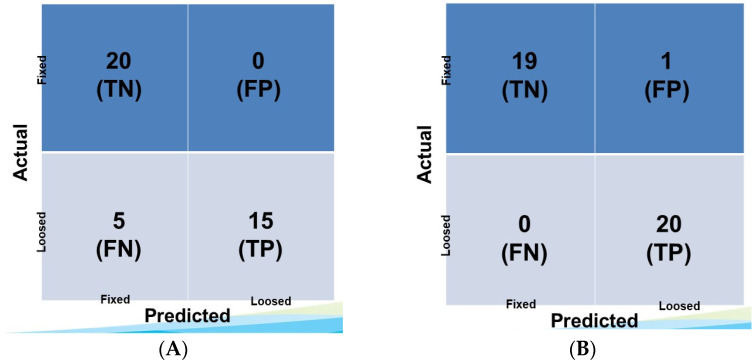
Confusion matrix for control and loosened TKA implants using (**A**) transfer learning model 1 and (**B**) transfer learning model 2.

**Figure 6 bioengineering-10-00632-f006:**
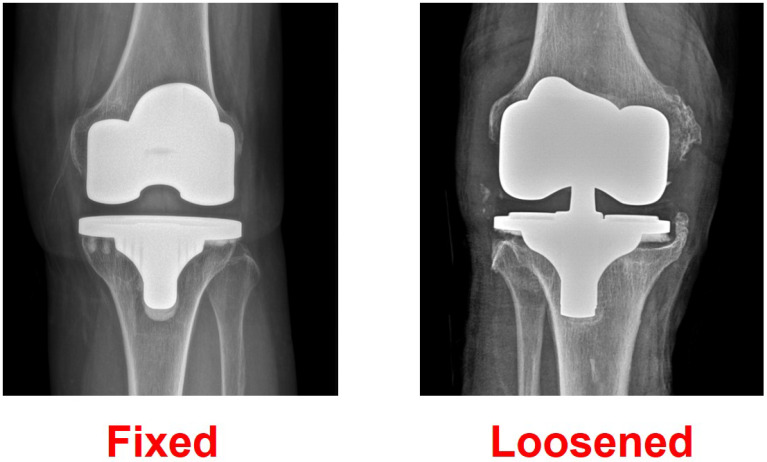
Prediction of fixed and loosened total knee arthroplasty implants.

**Table 1 bioengineering-10-00632-t001:** Patient characteristics before and after propensity score matching (PSM).

	Fixed (*n* = 399)	Loosened (*n* = 100)	*p*-Value
Demographics before PSM			
Age (years) *	69.7 ± 6.8	70.4 ± 8.3	0.430
Gender (female, %)	353 (88.5%)	80 (80.0%)	0.032
BMI (kg/m^2^)	26.1 ± 3.4	26.3 ± 3.4	0.430
Operation side (left, %)	203 (50.9%)	37 (37.0%)	0.013
ASA grade			0.073
1	45 (11.3%)	8 (8.0%)	
2	347 (87.2%)	87 (87.0%)	
3	6 (1.5%)	5 (5.0%)	
Demographics after PSM			
Age (years) *	70.9 ± 6.7	70.4 ± 8.3	0.603
Gender (female, %)	80 (80.0%)	80 (80.0%)	1.000
BMI (kg/m^2^)	26.5 ± 3.6	26.3 ± 3.4	0.649
Operation side (left, %)	37 (37.0%)	37 (37.0%)	1.000
ASA grade			0.238
1	7 (7.0%)	8 (7.0%)	
2	92 (92.0%)	87 (87.0%)	
3	1 (1.0%)	5 (5.0%)	

* The values are presented as mean and standard deviation. BMI, body mass index; ASA, American Society of Anesthesiologists.

**Table 2 bioengineering-10-00632-t002:** Image-based machine learning model performance using the test set.

Performance Criteria	Transfer Learning Model 1	Transfer Learning Model 2
Accuracy	87.5%	97.5%
Sensitivity	75.0%	100%
Specificity	100%	95.0%
Positive predictive value	100%	95.2%
Negative predictive value	80.0%	100%

## Data Availability

The data presented in this study are available in the main article.

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
