# Peer review of "Machine Learning for Detecting Total Knee Arthroplasty Implant Loosening on Plain Radiographs"

_bioengineering, 2023, doi:10.3390/bioengineering10060632_

Round 1

Reviewer 1 Report

Interesting article on a little covered topic.

The main question addressed by the research was machine learning for detecting total knee arthroplasty implant loosening on plain radiographs.

There are few related articles.

Well written abstract.

Introduction: well written, extensive literature review, clearly defined research objectives.

Materials and methods clearly and extensively presented.

Clear, legible and nice tables and figures.

The discussions should be improved by adding more arguments about the desirability of this program and study.

The references are appropriate.

My comments:

Please correct punctuation and stylistic errors throughout the text.

In the discussion and limitations, it should be described that the program allowed the recognition of already confirmed loosening of endoprostheses. However, it should detect not yet recognized loosening. Is there really a need for a program to recognize already detected loosening - if doctors recognize it earlier?

Interesting article on a little covered topic.

The main question addressed by the research was machine learning for detecting total knee arthroplasty implant loosening on plain radiographs.

There are few related articles.

Well written abstract.

Introduction: well written, extensive literature review, clearly defined research objectives.

Materials and methods clearly and extensively presented.

Clear, legible and nice tables and figures.

The discussions should be improved by adding more arguments about the desirability of this program and study.

The references are appropriate.

My comments:

Please correct punctuation and stylistic errors throughout the text.

In the discussion and limitations, it should be described that the program allowed the recognition of already confirmed loosening of endoprostheses. However, it should detect not yet recognized loosening. Is there really a need for a program to recognize already detected loosening - if doctors recognize it earlier?

Author Response

Interesting article on a little covered topic.

 â–¶Thank you for your comments.

The main question addressed by the research was machine learning for detecting total knee arthroplasty implant loosening on plain radiographs.

â–¶We thank the reviewer for his/her valuable time and we agree with this succinct summary of our study.

There are few related articles.

â–¶Thank you for your comments.

Well written abstract.

â–¶Thank you for your comments.

Introduction: well written, extensive literature review, clearly defined research objectives.

â–¶Thank you for your comments.

Materials and methods clearly and extensively presented.

â–¶Thank you for your comments.

Clear, legible and nice tables and figures.

â–¶Thank you for your comments.

The discussions should be improved by adding more arguments about the desirability of this program and study.

â–¶Thank you for your comments. We totally agreed with your opinion. This study has a limitation in that the model performance cannot be judged for those whose loosening has not yet been recognized because the model only judged the confirmed loosening of TKA implant on plain radiograph. Therefore, it is questionable whether our model would be effective for detecting loosening during early stages. It is thought that it can be the basis for developing a model that can recognize loosening at an early stage. A section on this issue has been added to the discussion of limitations. (Lines 241-245)

The references are appropriate.

â–¶Thank you for your comments.

My comments:

Please correct punctuation and stylistic errors throughout the text.

 â–¶Thank you for your comments. We corrected punctuation and stylistic errors throughout the revised manuscript.

In the discussion and limitations, it should be described that the program allowed the recognition of already confirmed loosening of endoprostheses. However, it should detect not yet recognized loosening. Is there really a need for a program to recognize already detected loosening - if doctors recognize it earlier?

â–¶Thank you for your comments. We totally agreed with your opinion. This study has a limitation in that the model performance cannot be judged for those whose loosening has not yet been recognized because the model only judged the confirmed loosening of TKA implant on plain radiograph. Therefore, it is questionable whether our model would be effective for detecting loosening during early stages. It is thought that it can be the basis for developing a model that can recognize loosening at an early stage. A section on this issue has been added to the discussion of limitations. (Lines 241-245)

Reviewer 2 Report

The paper describes how a popular image classification AI technique (CNN+VGG19) has been adapted for a particular application - detecting knee arthroplasty implant loosening from plain radiographs - and has been demonstrated with a high degree of success. While how adaptation or modification was done has been described in some detail, the following are not as clear, and should be added/improved in my opinion:  

(1)  In Introduction (Lines 51/52), it’s stated that “Machine learning can be used to detect prosthetic loosening after arthroplasty in orthopedic surgery but has many limitations in this application”. What are these limitations and which – if any – your research paper is trying to address?

(2)  Lines 70/71: 224x224 seems to be the default image size for VGG19. Is such image size sufficient for your subject? You mentioned “resizing”, was it scaling the original images up or down?

(3)  How was the input images prepared? Manually cutting out the relevant part, or using another (possible AI based) image processing software to automatically extracting the relevant part from plain radiographs?

(4)  Line 71: “an augmentation technique” was used, but not described in any detail to allow the reader to understand what’s exactly been done.

(5)  Fig.3: do “accuracy” and “loss” add up to 1? If so, one of the plots should suffice. If not (and at least for “test” it looks like they don’t exactly add up to 1), why not? Ditto for Fig.4.

(6)  Fig.6: the images look higher resolution than 224x244. I wonder what 224x224 images look like – how much detail are lost?

(7)  In Conclusions (Lines 257/258), its potential application is speculated (i.e., “may be utilized as an auxiliary tool for diagnosing…”). Some of the limitations have already been mentioned and discussed in Section 4. Still it would be helpful here to list and rank the main challenges still remain before this technique can be deployed for real. In particular, comments relating to Points (2) and (3) above.  

Last point: I don’t know if it’s this journal’s format requirement, but having citation after sentence stop just doesn’t look right to me. Example “…model. [25,28] However…”.  

Author Response

The paper describes how a popular image classification AI technique (CNN+VGG19) has been adapted for a particular application - detecting knee arthroplasty implant loosening from plain radiographs - and has been demonstrated with a high degree of success. While how adaptation or modification was done has been described in some detail, the following are not as clear, and should be added/improved in my opinion:  

â–¶We thank the reviewer for his/her valuable time and we agree with this succinct summary of our study.

(1)  In Introduction (Lines 51/52), it’s stated that “Machine learning can be used to detect prosthetic loosening after arthroplasty in orthopedic surgery but has many limitations in this application”. What are these limitations and which – if any – your research paper is trying to address?

 â–¶Thank you for your comments. Although many studies on image evaluation using machine learning are in progress, it is evaluated that the studies on models evaluating loosening after TKA are limited because there are few and insufficient studies. Due to a misunderstanding, what should have been expressed as 'the studies are still lacking' was expressed as 'there are many limitations'. It has been changed to “the studies are still lacking”. We changed the sentence in the revised manuscript. (Lines 51-52)

(2)  Lines 70/71: 224x224 seems to be the default image size for VGG19. Is such image size sufficient for your subject? You mentioned “resizing”, was it scaling the original images up or down?

 â–¶Thank you for your comments. Since the size of the original image was different for each image, it was modified and there was no significant change from the original image. Despite scaling to 224x224, the accuracy of the transfer learning model was high. We added this issue in the revised manuscript. (Lines 73-74)

(3)  How was the input images prepared? Manually cutting out the relevant part, or using another (possible AI based) image processing software to automatically extracting the relevant part from plain radiographs?

 â–¶Thank you for your comments. The image was separated using the cropping technique through numpy in python by focusing the image around the implant-bone interface, which was recognized as the most important position, to better confirm loosening around the implant. We added this issue in the revised manuscript. (Lines 69-72)

(4)  Line 71: “an augmentation technique” was used, but not described in any detail to allow the reader to understand what’s exactly been done.

â–¶Thank you for your comments. The augmentation technique was implemented using rotation, width shift, height shift, zoom, and flip technique. We added this issue in the revised manuscript. (Lines 76-78)

(5)  Fig.3: do “accuracy” and “loss” add up to 1? If so, one of the plots should suffice. If not (and at least for “test” it looks like they don’t exactly add up to 1), why not? Ditto for Fig.4.

â–¶Thank you for your comments. Loss and Accuracy have nothing to do with each other. Accuracy can be viewed as the number of prediction errors for the entire data. Simply put, it is how many of the total data were correct. Loss is the difference (distance or error) between the actual correct answer and the value predicted by the model. That is, the larger the loss, the larger the error for the data. In simple terms, it can be seen as how much error you made if you predicted incorrectly. (Figures 3 and 4)

(6)  Fig.6: the images look higher resolution than 224x244. I wonder what 224x224 images look like – how much detail are lost?

â–¶Thank you for your comments. Although there’s a loss, we changed it to the form of 224x224 to put it in the model. If the data is imbalanced, it is difficult to secure practicality and objectivity even if the model performance is excellent. Since the pixels are almost identical, there is little loss of data. We changed fig 6 to a radiograph with 224x224 pixels in the revised manuscript. (Figure 6)

(7)  In Conclusions (Lines 257/258), its potential application is speculated (i.e., “may be utilized as an auxiliary tool for diagnosing…”). Some of the limitations have already been mentioned and discussed in Section 4. Still it would be helpful here to list and rank the main challenges still remain before this technique can be deployed for real. In particular, comments relating to Points (2) and (3) above.  

â–¶Thank you for your comments. This study has a limitation in that the model performance cannot be judged for those whose loosening has not yet been recognized because the model only judged the confirmed loosening of TKA implant on plain radiograph. Therefore, it is questionable whether our model would be effective for detecting loosening during early stages. It is thought that it can be the basis for developing a model that can recognize loosening at an early stage. In addition, preprocessing, which resizes and crops images to substitute for transfer learning models, can reduce image resolution, which can have a significant impact on the performance of the model. We added this issue in the limitation of the revised manuscript. The last sentence of the conclusion is misleading and has been changed as follows. This method showed the possibility of being used as an auxiliary tool for diagnosing prosthetic loosening of TKA implants, but it is judged that additional research is needed. (Lines 240-247, 266-267)

Last point: I don’t know if it’s this journal’s format requirement, but having citation after sentence stop just doesn’t look right to me. Example “…model. [25,28] However…”.  

â–¶Thank you for your comments. We changed the position of citation to fit the format of the journal.

Round 2

Reviewer 1 Report

The authors have made the suggested changes.

Accept in present form.